State of biodiversity documentation in the Philippines: Metadata gaps, taxonomic biases, and spatial biases in the DNA barcode data of animal and plant taxa in the context of species occurrence data

Berba Carmela Maria P.
http://orcid.org/0000-0002-1329-4285 Matias Ambrocio Melvin A. aamatias@up.edu.ph
Institute of Biology, University of the Philippines Diliman , Quezon City, National Capital Region , Philippines
Sosa Victoria
Electronic publication date: 2022 Mar 21
Publication date: 2022
Volume: 10
Electronic Location ID: e13146
Received 2021 Sep 23; Accepted 2022 Mar 1
Copyright: © 2022 Berba and Matias
Copyright year: 2022
Copyright holder: Berba and Matias
License: This is an open access article distributed under the terms of the Creative Commons Attribution License, which permits unrestricted use, distribution, reproduction and adaptation in any medium and for any purpose provided that it is properly attributed. For attribution, the original author(s), title, publication source (PeerJ) and either DOI or URL of the article must be cited.
License URL: https://creativecommons.org/licenses/by/4.0/

Keywords: Comparative analysis, Conservation, Genetic diversity, Online biodiversity database, Sampling biases, Spatial analysis, Species diversity

Funding: The authors received no funding for this work.

==============================
Anthropogenic changes in the natural environment have led to alarming rates of biodiversity loss, resulting in a more urgent need for conservation. Although there is an increasing cognizance of the importance of incorporating biodiversity data into conservation, the accuracy of the inferences generated from these records can be highly impacted by gaps and biases in the data. Because of the Philippines’ status as a biodiversity hotspot, the assessment of potential gaps and biases in biodiversity documentation in the country can be a critical step in the identification of priority research areas for conservation applications. In this study, we systematically assessed biodiversity data on animal and plant taxa found in the Philippines by examining the extent of metadata gaps, taxonomic biases, and spatial biases in DNA barcode data while using species occurrence data as a backdrop of the ‘Philippines’ biodiversity. These barcode and species occurrence datasets were obtained from public databases, namely: GenBank, Barcode of Life Data System and Global Biodiversity Information Facility. We found that much of the barcode data had missing information on either records and publishing, geolocation, or taxonomic metadata, which consequently, can limit the usability of barcode data for further analyses. We also observed that the amount of barcode data can be directly associated with the amount of species occurrence data available for a particular taxonomic group and location–highlighting the potential sampling biases in the barcode data. While the majority of barcode data came from foreign institutions, there has been an increase in local efforts in recent decades. However, much of the contribution to biodiversity documentation only come from institutions based in Luzon.

Introduction

Biodiversity is the product of the interactions between many physical and biological processes across time (Boero & Bonsdorff, 2007; van der Plas, 2019). Unfortunately, recent anthropogenic activities have significantly impacted biodiversity resulting in its rapid decline (Halpern et al., 2008, 2015). If left unabated, this alarming biodiversity loss can potentially impair the capacity of ecosystems to support and sustain life over time (Ayyad, 2003; Butchart et al., 2010; Cardinale et al., 2012; Reich et al., 2012; Worm et al., 2006). Due to these anthropogenic impacts on biodiversity, conservation efforts have been implemented to mitigate biodiversity loss and to promote the recovery of affected ecosystems and species. These initiatives include prioritization and management of key areas that best represent biodiversity or the processes (i.e., ecological and evolutionary) sustaining it (Beger et al., 2014; Herrick, Schuman & Rango, 2006; Hoffmann & Sgró, 2011; Moritz, 2002; Richardson & Whittaker, 2010; Selig et al., 2014; Sgrò, Lowe & Hoffmann, 2011). However, efforts to conserve biodiversity could potentially be ineffective, or even counterproductive, if there is a lack of understanding of the fundamental processes underlying biodiversity (e.g., Hoveka et al., 2020; Santangeli et al., 2013). Thus, an understanding of biodiversity and the processes underpinning it is necessary in order to improve the efficacy of conservation efforts.

Because biodiversity is organized at different levels (i.e., ecosystems, species, and genes), making inferences about biodiversity-generating processes that are relevant to conservation will require documentation and analysis of biodiversity at various levels (Laikre et al., 2010; Purvis & Hector, 2000; Sarkar & Margules, 2002). Although significant progress has been made regarding biodiversity documentation, there has always been a tendency for biodiversity data to be spatially and taxonomically biased. These biases are often in contrast with the natural patterns and distribution of biodiversity (Titley, Snaddon & Turner, 2017; Troudet et al., 2017). For example, globally, biodiversity documentation is biased towards developed countries within temperate regions despite the tropical regions being relatively more diverse (Meyer et al., 2015; Newbold, 2010; Titley, Snaddon & Turner, 2017). At regional scales, spatial bias is also prominent primarily because many biodiversity documentations are results of scientific research focused on answering specific questions. Consequently, sampling is associated with certain geographical features related to the research question (e.g., near or within protected areas). This bias potentially leads to the under-representation of many key habitats in biodiversity documentation (Fisher-Phelps et al., 2017; Newbold, 2010). Current knowledge on biodiversity is further biased towards more charismatic organisms (i.e., mostly plants and vertebrates) leaving significantly more diverse taxonomic groups, such as invertebrates, understudied (Titley, Snaddon & Turner, 2017; Troudet et al., 2017). Overall, the extent of biases in biodiversity documentation reflects the insufficient data in many regions and taxa, which are likely due to limited research topics brought by various historical, social, economic, and practical factors (dos Santos et al., 2020; Troudet et al., 2017).

The various spatial and taxonomic biases in biodiversity data can potentially affect key inferences about biodiversity-related processes (e.g., Keyse et al., 2014; Matias & Riginos, 2018). Because these inferences are explicitly being incorporated in conservation, these biases can potentially lead to poorly-advised decisions that may contribute to biodiversity decline. Moreover, conservation entails costs at various stages of its implementation (i.e., opportunity, acquisition, management, and maintenance), and providing for this cost involves allocation of highly-constrained resources such as time and money (Margules & Pressey, 2000; Possingham & Wilson, 2005). Thus, mitigating the impact of these biases can benefit conservation efforts by making them cost-effective in the use of those valuable resources, particularly for countries where such resources are limited but where conservation is in demand.

One example of countries that will benefit greatly from cost-effective conservation efforts is the Philippines. This country is a tropical developing country that has been considered as one of 17 megadiverse nations worldwide (Mittermeler & Mittermeler, 1997), largely due to its rich diversity and endemism. It has been estimated that there are more than 38,000 species of vertebrates and invertebrates in the country (Catibog-Shinha & Heaney, 2006)–a likely conservative number given the variability in estimates across groups. For example, as new species are being discovered, some reports have predicted that Philippine arthropod species would eventually reach 50,000 to 100,000 in number (Gapud, 2002). For plant taxa, around 14,000 species are found in the Philippines (Madulid, 1985 as cited in Lagunzad, Co & Navarro (2002)) along with 35 of 54 mangrove species (Tomlinson, 1986 as cited in Primavera (2002)), more than 1,000 seaweed species (Fortes, 2002a), and 16 seagrass species (M. Fortes, 1986 as cited in Fortes (2002b)). Among the animal and plant species that have been described so far, more than half of them are said to be endemic to the Philippines (Ong, 2002).

Despite the number of species that have already been described in the Philippines, there are still a lot of uncertainties regarding the estimated biodiversity in the country. Moreover, there is also growing threats on the local environment as the Philippines becomes one of the “hottest” biodiversity hotspots in the world due to the amount and rate of loss and degradation in various habitats (Halpern et al., 2015; Harvey et al., 2020; Myers et al., 2000). These threats to biodiversity have increased the need for conservation. Yet, the gaps in biodiversity documentation in the country can potentially constrain these efforts. Addressing this problem will require the identification of biases present in biodiversity records. Thus, a comprehensive and systematic assessment of the current biodiversity data is needed to ensure the efficacy of future conservation efforts based on such information.

Previous works that have examined biases and gaps in biodiversity data have utilized publications collected from search engines such as the Web of Science (dos Santos et al., 2020; Titley, Snaddon & Turner, 2017) or certain biodiversity records obtained from public databases. For example, DNA barcode data from GenBank identified through published work has been used to examine the extent of DNA barcoding in the Philippines (Fontanilla et al., 2014). Similarly, for many works, species occurrence data from the Global Biodiversity Information Facility (GBIF) is used (Fisher-Phelps et al., 2017; Meyer et al., 2015; Oliveira et al., 2016; Troudet et al., 2017). Importantly, in these previous examinations, species occurrence and DNA barcode data are typically examined separately for biases and gaps. However, given that the components of biodiversity and its underlying processes are fundamentally intertwined (e.g., genetic data shedding light on cryptic species diversity), it becomes critical that species and genetic data are examined side by side. This approach can potentially help identify common patterns of biases and gaps in the documentation of biodiversity at both levels.

In this study, public databases are leveraged to systematically examine potential gaps and biases present in current records and gain a better understanding of the state of biodiversity documentation in the country. The study specifically focuses on public biodiversity data of animal and plant taxa found in the Philippines that are accessible in three online databases, namely: the Global Biodiversity Information Facility (GBIF), GenBank, and Barcode of Life Data System (BOLD). These databases represent large repositories of biodiversity records that are widely used among the scientific community–as well as citizen scientists mainly in the case of GBIF (Petersen et al., 2021)–to publish data. Because these datasets are readily accessible, they represent records more frequently processed and analyzed to generate inferences for policymaking and conservation planning (Ball-Damerow et al., 2019). Thus, examining biodiversity data from these databases will not only identify biases in the current data but can also mitigate the risks posed by these biases to conservation efforts. Although both species and genetic data will be utilized, the analyses in this study will mainly focus on the genetic data with species data serving as a background. Because species data from public database have prominent biases (some inherent with citizen science and its opportunistic nature of collection), its comparison with genetic data can potentially highlight biases in genetic data as well (Amano, Lamming & Sutherland, 2016; Petersen et al., 2021; Troudet et al., 2017). To systematically assess both datasets, species and genetic data are examined for the following: (1) metadata gaps in relation to the completeness of biodiversity records; (2) taxonomic biases at the species and genetic levels; and (3) spatial biases in terms of sampled locations and origin of leading contributors. These assessments are done to identify potential knowledge gaps present in Philippine biodiversity documentation. This approach is a key step in addressing biases to generate more accurate inferences and develop better strategies on how to move forward in future efforts in biodiversity documentation and conservation.

Materials and Methods

Collecting and parsing of biodiversity data

In examining Philippine biodiversity data, we limited our collection of data to three databases that are widely used and easily accessible. Thus, our study represents information that is likely to be used by many researchers or even policymakers. We obtained species occurrence data directly from the Global Biodiversity Information Facility (GBIF, https://www.gbif.org/) on October 18, 2020 (GBIF.org, 2020a, 2020b). The search was filtered by country (“Philippines”), occurrence states (“Present”), and taxonomic key (“Animalia” and “Plantae”). The barcode data was obtained directly from two separate databases, namely: GenBank (https://www.ncbi.nlm.nih.gov/genbank/) on November 1 and 3, 2020 and Barcode of Life Data System (BOLD, http://v4.boldsystems.org/) on November 3, 2020. In GenBank, searches were conducted using different sets of keywords to obtain barcode data based on the gene marker of interest. The gene markers actively searched for in GenBank were the following: cytochrome oxidase c subunit I (using the keywords, “COI OR co1 OR cox1 OR coxI OR cytochrome oxidase OR cytochrome c oxidase AND Philippines”); cytochrome b (using the keywords, “cytb OR cyt-b OR cyt b OR cytochrome b OR cytochrome-b AND Philippines”); ribulose-1,5-biphosphate carboxylase (using the keywords, “ribulose-1,5-bisphosphate carboxylase OR rbcl OR rubisco OR ribulose-bisphosphate carboxylase AND Philippines”); maturase K (using the keywords, “matk OR MaturaseK OR maturase K AND Philippines”); and lastly, internal transcribed spacer two (using the keywords, “internal transcribed spacer 2” OR ITS2 OR ITS AND Philippines”). Prior to downloading data from GenBank, the results of each search were filtered based on species to only include “Animals” and “Plants”. It is important to note that the data obtained may have included entries labelled as “unverified” since our searches were unfiltered for verification. In BOLD, several searches were conducted in the Public Data Portal system based on geography (keyword, “Philippines”) and taxonomy (using all taxonomic groups listed under animals and plants in BOLD’s Taxonomy Browse–http://v4.boldsystems.org/index.php/TaxBrowser_Home).

We mainly utilized the data.table R package (Dowle & Srinivasan, 2020) to manage and parse through the data we obtained. However, in the case of GenBank data, the downloaded data had to be processed into more readable files for each data entry. We used our own set of R functions–specifically made to parse through individual GenBank files–to pull out as much information as possible and organize it into a more workable data frame. We created seven functions that obtained the following information: (1) taxonomy of the specimen; (2) publishing author; (3) publishing institution; (4) year submitted; (5) metadata associated with the “source”; (6) gene marker; and (7) barcoding sequence (made available in github.com/dinmatias). We also conducted additional cleaning and fixing on the information pulled out from the GenBank files on BOLD cross-reference, taxonomy, publishing institution, gene marker, and sampling location. For the taxonomy information, we created a database derived from the unique species found in GBIF to obtain only the information on phylum/division, class, order, family, and genus while other taxonomic ranks were disregarded. To obtain the publishing institution, we manually parsed through the unique publishing entries and narrowed down the information to two columns that contained the name of the main institution involved (labelled as PublishingInstitution) as well as the country where it is based (labelled as PublishingCountry). For the BOLD data, an additional column was added for the country where the storing institution, copyright institution, and sequencing center are based. Some of the gene markers entries initially pulled out were unclear or vague due to the varying ways the information was laid out in the individual GenBank files and how the markers were named (e.g., full name or different abbreviations). For these reasons, these entries were manually parsed to standardize the names of the gene markers used. While the coordinate entries for the sampling information required minimal cleaning, the descriptive information on the locality where the specimen was sampled required intensive manual parsing. This editing was done not only for GenBank data but also for BOLD data to obtain the specific information on province, municipality, and/or barangay based on a location database derived from the Philippine Standard Geographic Code (PSGC) (Philippine Statistics Authority, 2020). During the parsing and cleaning process, sampling information was categorized based on the kind of issues encountered during the parsing (if any) that made them vague or inconclusive (see Table S1). Moreover, the descriptive information provided for the sampling locality in the GBIF data was parsed through and cleaned such that it was organized into province, municipality, and/or barangay.

After parsing and cleaning the data, we obtained the subsets of the main datasets containing the metadata associated with the following categories: records (i.e., entry ID and collection date), taxonomy (i.e., phylum/division, class, order, family, genus, species), geolocation (i.e., coordinates and administrative units where the specimen was sampled), and publication (i.e., submission date, publishing institution, and country) (see Table S2). For taxonomy, we recognize that there are differences between animal and plant taxonomy, particularly with regards to the taxonomic ranks lower than kingdom–e.g., phylum for animal taxa and division for plant taxa. However, phylum and division were placed in the same taxonomic metadata in the species and genetic databases we collected from–generally being categorized as “phylum”. Hence, in this study, phylum and division were treated as one classification in the analyses. For the downstream analyses, the GenBank and BOLD datasets were combined into one barcode dataset after selecting the metadata of interest. In combining these two datasets, we ensured that the columns (variables) were analogous between the two databases. We further filtered our two main working datasets (i.e., species occurrence and barcode data) by excluding the following entries: duplicates in barcode data based on accession number; gene markers that were not part of the five markers actively searched for; barcode specimen sampled from foreign countries; and species occurrence and barcode data on Homo sapiens and H. luzonensis. Additionally, a substantial number of barcode records with missing information on the country of collection was observed despite having filtered the searches based on geography. Because this number was substantial, two sets of analyses were conducted: (1) one where NA was excluded and (2) another where NA was included in the dataset. While it is likely that the latter approach may have included a few sequences that are not actually from the Philippines, the results were generally the same between the two sets of analyses. Thus, the results from the latter approach were mainly presented.

Examining for metadata gaps

To assess the completeness of the metadata associated with the barcode data, we quantified the number of records with missing information on the following categories: publication and records, sampling location, and taxonomy. In the publication and records category, the number of records that lacked information on the copyright institution, collection year, and submission year were counted. In the sampling location category, we counted the number of records that lacked coordinates (i.e., latitude and longitude) and within this data subset, the proportion of records with (or without) additional information on the sampling locality was examined. Additionally, we determined the frequency of each kind of issue encountered while manually parsing through the descriptive information on the sampling locality–with those having more than one issue being categorized as “mixed”.

In the taxonomy category, we first assessed the entries that had information on the species level but lacked information on one or more higher taxonomic ranks. Here, the original entries for the species information that included the keywords, “sp.” and “gen.” were marked as NA since the true species identity was not provided. For records with identified species but incomplete taxonomic data, we attempted to fill in the missing entries using the same database we derived from the taxonomy of unique species in GBIF. Because barcode data is mainly used as a reference in “species identification”, the use of sequences that are not identified to species level is not maximized. Hence, to identify and examine the taxonomic groups with barcode data with low species identification, we plotted the percent of identified species in barcode data against the percent of species with available barcode records that are represented in species occurrence data. This was done separately for animal and plant records at the phylum/division, class, order, and family levels.

Examining for taxonomic biases

To compare the extent of species and genetic documentation among taxonomic groups, we plotted the number of available records per taxon in barcode data against that of species occurrence data. The data was first transformed using logarithmic function prior to plotting. Similar to the previous section on taxonomic metadata gaps, this was done separately for animal and plant records at the phylum/division, class, order, and family levels. Additionally, quantiles–specifically, the 5th and 95th percentile–of both datasets were incorporated in the plots to highlight taxonomic groups on the extreme 10% of the distribution of these two variables. Here, the occurrence record was used as a measure of the commonness of a taxonomic group in examining how well commonly recorded taxonomic groups are being barcoded.

Examining for spatial biases

To assess the sampling distribution of barcode and species data, we first obtained shapefiles of the Philippine administrative boundaries, specifically, the Philippines-Subnational Administrative shapefile (https://data.humdata.org/dataset/philippines-administrative-levels-0-to-3). Using this database, the province information of a given coordinate entry was determined based on which defined boundaries of the administrative level 2 (i.e., province boundary) it falls under. In the case of marine specimens with coordinates that do not fall within any province boundary (because the boundary is based on land), the nearest province to them was assigned as their province information. The nearest province was determined by first identifying the “centroid” of each province and then measuring the distance of a data point to the centroid. The province with the shortest distance from the data point was assigned as its province. For records without any coordinates, only records with information on the province where the specimen was sampled were included. These filtered datasets were then used to generate separate heatmaps for the sampling distribution of barcode and species occurrence data. Moreover, we also plotted the number of records per province in barcode data against that of species occurrence data, with the data transformed logarithmically prior to plotting and the 5th and 95th percentiles incorporated.

To examine the distribution of global contribution to Philippine barcode data, we focused on the countries where the institutions that submitted or, in the case of BOLD, held the copyright to the image data are from (i.e., copyright_institutions). Another metadata column in BOLD that was considered to be examined for contribution was the institute that served as the storage place of the voucher (i.e., institution_storing); however, the entries of the two columns were generally the same. We quantified the number of barcode records published per country and visualized their spatial distribution through the wrld_simpl shapefile from the maptools R package (Bivand & Lewin-Koh, 2021). Additionally, the contribution of local and foreign efforts in generation barcodes across time was compared. For this comparison, barcode records were categorized as contributed by either “Foreign” or “Philippines” based on the copyright country. This information was summarized into two plots showcasing the barcoding activity through time in terms of year of collection (starting from the 1990s) and year of submission/publication (starting from the 2000s). Note that we presented the barcoding activity across the year by “smoothened” curved obtained through local regression (i.e., loess regression).

We then examined the contribution to barcode data at the national level–meaning different institutes based in the Philippines. For each barcode record, we assigned the “processing center” (i.e., region where the institute holding the copyright is located) and “region sampled” (i.e., region where the specimen was collected). The total number of barcode records generated by each “processing center” from a specific “region sampled” was used as its contribution per “region sampled”. The local contribution data was then summarized via a correlation matrix heatmap, which plotted the region of sampling against the region of local institutions. In this matrix, the regions were sorted according to their proximity to provide spatial context. We utilized the following R packages to conduct our spatial analyses: sp (Bivand, Pebesma & Gomez-Rubio, 2013; Pebesma & Bivand, 2005), raster (Hijmans, 2020), rgdal (Bivand, Keitt & Rowlingson, 2021), and RColorBrewer (Neuwirth, 2014).

Results

Initial processing of biodiversity data

From the initial database searches conducted in late October to early November 2020, a total of 31,163 barcode records–18,094 from GenBank and 13,069 from BOLD–and 1,557,709 species occurrence records were retrieved. Upon parsing through the raw datasets, duplicates, unwanted gene markers, and foreign samples in the barcode data as well as records involving H. sapiens and H. luzonensis in both barcode and species data were excluded. This initial filtering resulted in 20,482 barcode (16,719 excluding NA entries for country sampled) and 1,557,374 species records available for downstream analyses. For the barcode data, the majority of the records obtained are based on the COI gene marker (see Fig. 1A). This may be linked to the significantly higher number of animal records analyzed in comparison to the number of plant records (a trend also observed in the available species occurrence data, see Table S3) since gene markers are often utilized for certain organisms (e.g., COI for animals then rbcL and matK for plants).

Figure 1 Summary of barcode records associated with specific gene markers and issues encountered while manually parsing through the descriptive information on sampling locality.

For graph A, the genetic summary of the available barcode records focuses on the gene markers of interest used in the examination for metadata gaps, taxonomic biases, and spatial biases in DNA barcode data on animal and plant taxa sampled in the Philippines were the following: cytochrome b (CYTB), cytochrome oxidase c subunit I (COI), internal transcribed spacer 2 (ITS2), ribulose-1,5-biphosphate carboxylase (rbcL), and maturase K (matK). For graph B, the geolocation issues resulted in the descriptions of the sampling location (particularly in terms of administrative units) being unclear or in some cases, inconclusive. The categories include misspelled (incorrect spelling), none (no major issue), mixed (more than one issue), unspecified (somewhat informative but still vague), unknown (completely not informative), multiple (provided more than one location), and mismatch (discrepancies between the administrative units provided). This dataset includes the records with NA entries for country sampled (for A and B) and those that had additional information on the geolocation other than the coordinates (for B only).

Metadata gaps in Philippine barcode data

Most of the barcode data used in the analyses were observed to have incomplete information in one or more categories of metadata. For the gaps in the records and publishing metadata, among the barcode data, 72.52% lacked information on the year of collection (66.73% excluding NA entries for country sampled), 22.01% on the year of submission (26.93% excluding NA entries for country sampled), and 18.51% on the publishing or copyright institution (22.64% excluding NA entries for country sampled). For the gaps in the geolocation metadata, approximately 65.78% had no coordinates (58.10% excluding NA entries for country sampled) and within that subset of data, more than half lacked any additional descriptive information on the sampling locality such as province, municipality, and barangay. Overall, 46.68% of barcode records lacked any kind of metadata on the sampling location (34.69% excluding NA entries for country sampled). Records that did have metadata on the sampling locality in terms of administrative units were relatively difficult to parse through. Majority of them were vague in varying degrees depending on the kind of major issue encountered–with most being unspecified (see Fig. 1B). Additionally, there were several records wherein “Philippines” was indicated as the country sampled but upon further inspection of the description of the specific locality sampled, a mismatch was found. Such entries were labelled as foreign and excluded from the analyses.

For the gaps in the taxonomic information, 3,793 records had no information on the specific group in one or more taxonomic ranks despite the specimen being identified at the species level. Using a taxonomic database derived from the species occurrence data, these gaps were filled in at the phylum/division, class, order, and family levels, narrowing down the number to 706 records with incomplete taxonomic information. The proportion of identified animal and plant species was also assessed in relation to the proportion of barcoded species per taxon at a specific taxonomic rank–namely, phylum/division, class, order, and family (see Fig. 2). At the phylum/division level, most of the taxa exhibited more than 50% species identification except for Annelida and Rotifera (see Fig. 2A). However, at lower taxonomic ranks, there were more taxa that had the majority (more than 50%) of their records unidentified at the species level (see Figs. 2B to 2D). Moreover, while more taxa were being sampled, the rate at which these groups were barcoded remains relatively low. Evidently, only a few groups exhibited a high percentage of identified and barcoded species. It is important to note, however, that the identity of the species was based on the information provided by the contributors who published the barcode records. It was not verified if the species identities matched with the barcode sequences associated with them. Additionally, in evaluating the proportion of barcoded species at the order and family level, several taxa returned an undefined value (NaN). These were likely the result of the absence of species occurrence records associated with those taxa despite having barcode records available. There were eight (8) orders resulting in NaN, labelled as the following: “Labriformes”, “Ovalentaria”, “Gobiiformes”, “Trachiniformes”, “Pristiformes”, “Pulmonata”, “Vetigastropoda”, and “Sebdeniales”. On the other hand, there were five (5) resulting NaN families, labelled as: “Pentanchidae”, “Chilodontidae_gas”, “Choristellidae”, “Sebdeniaceae”, and “Areschougiaceae”.

Figure 2 Relationship between the percentage of barcode records identified at the species level and the proportion of documented species (represented in species occurrence data) that currently have DNA barcode data available.

This relationship was evaluated for each known animal (orange) and plant (green) taxonomic group represented in the Philippine barcode data at the phylum/division (A), class (B), order (C), and family (D) levels. This dataset includes the records with NA entries for country sampled.

Taxonomic biases in Philippine barcode data

Examination of the taxonomic distribution of records collected revealed a general increasing trend between the amount of barcode and species occurrence data for a particular taxon (see Fig. 3). At the phylum/division level, the group with the highest record in both barcode and species data was Chordata and accompanying it in the areas of either high genetic data or high species data were Arthropoda, Mollusca, and Tracheophyta (see Fig. 3A). On the other hand, the groups that had particularly low biodiversity records, particularly in terms of barcode data, were Rotifera, Ctenophora, and Marchantiophtya. There were several taxa that had species occurrence data but lacked barcode data, namely: Anthocerotophyta, Brachiopoda, Bryozoa, Cephalorhyncha, Chaetognatha, Charophyta, Entoprocta, Hemichordata, Nematomorpha, Phoronida, Sipuncula, and Xenacoelomorpha. Assessing the trends further down the taxonomic hierarchy, it could be observed that while more groups had been sampled in terms of species occurrence, many of them had little to no barcode records available (see Figs. 3B to 3D). Furthermore, groups that remained at or above the 95th percentile of genetic and species data at the class, order, and family levels mostly belonged to Phylum Chordata.

Figure 3 Relationship between the amount of genetic and species data associated with each known animal and plant taxonomic group represented in the Philippine biodiversity data at different taxonomic levels.

This relationship was evaluated for each known animal (orange) and plant (green) taxonomic group represented in the Philippine barcode data at the phylum/division (A), class (B), order (C), and family (D) levels. Values were transformed logarithmically prior to plotting however, taxa with zero (0) records in either genetic or species data were assigned the value of negative one (−1). Dashed lines represent the 5th and 95th percentiles for genetic (horizontal) and species (vertical) data. This dataset includes the records with NA entries for country sampled.

Spatial biases in Philippine barcode data

Examination of the spatial distribution of records obtained showed a high similarity between the sampling distributions of barcode and species occurrence data, particularly in terms of the provinces wherein sampling was most and least concentrated (see Figs. 4A and 4B). In both genetic and species data, the province that had been relatively more sampled (above the 95th percentile) was Palawan. These similarities in sampling distribution meant that the amount of barcode data could be directly related to the amount of species occurrence records sampled per province (see Fig. 4C)–similar to the previous section on taxonomic bias. Furthermore, several provinces were observed to fall under the 95th percentile of either dataset. For barcode data, in particular, the provinces with the highest records (above 95th percentile) were Siquijor, Cavite, Bohol, Aurora, and Palawan while the ones with the lowest records (below 5th percentile) were Tarlac, Basilan, Maguindanao, Zamboanga Sibugay, and Northern Samar.

Figure 4 Maps of the sampling distribution of barcode and species occurrence data on animal and plant taxa across the Philippines and the relationship between the two datasets in terms of province.

For both maps (A – barcode data and B – species occurrence data), records on marine specimens were assigned to a specific province based on which corresponding centroid has the shortest distance from the given sampling coordinates (if available). Also, values presented in the maps represent the number of records in the thousands. In the scatter plot (C), values were transformed logarithmically and provinces with zero (0) records in either genetic or species data were assigned the value of negative one (−1). Dashed lines represent the 5th and 95th percentiles for genetic (horizontal) and species (vertical) data. The barcode dataset includes the records with NA entries for country sampled.

Examination of the institutions contributing to the barcode data revealed that in provinces where barcode sampling was most concentrated, the majority of the records were generated by foreign institutions. A notable exemption was Pangasinan, the seventh most sampled location in terms of barcoding data, majority of which were contributed by local institutes (~70.42% of the records). A similar trend of high contribution by foreign institutions to barcoding was observed when all barcode data were examined. While the Philippines had the most contribution to its barcode records compared with other countries (see Fig. 5A), a comparison of the foreign and local contributions showed that the Philippines had contributed only about 30.00% of the overall barcode data on Philippine animal and plant biodiversity.

Figure 5 Map of the distribution of barcode data on Philippine animal and plant biodiversity contributed by different countries across the world and their contribution to documenting efforts across the years.

For map A, contribution was based on the institution that holds the copyright to the image associated with the records while for the graphs, it was based on the collection of samples, starting from the 1990s (B) and submission of barcode data, starting from the 2000s (C) by foreign countries (violet) and the Philippines (red). Trendlines in the graphs represent the average, “best” fitted line. This dataset includes the records with NA entries for country sampled.

When foreign and local contribution of barcode data were examined across time–specifically, the time of collection and submission–it was revealed that the Philippines had increasingly collected and submitted more records by 2005. Moreover, at some point, the Philippines had even surpassed the activity of foreign institutions (see Figs. 5B and 5C). Additionally, though not represented in Fig. 5B, many of the specimens used by foreign institutes in generating barcode data had been collected before the 1990s, even as far back as 1915, highlighting the importance of sample preservation in documenting not only species but potentially genetic diversity as well.

Within the Philippines, there was a substantial discrepancy in contributions of local institutions to barcode data (see Fig. 6). When the regions of barcode-generating institutions (termed as the “Processing Center”) were compared with regions where sampling was conducted, it was apparent that only six of seventeen regions were able to generate barcode data for their local biodiversity (diagonals in Fig. 6). Furthermore, most local contributions were from institutions found in the regions of Metro Manila and Central Luzon, and these institutes sampled the most either within their local area or in nearby regions, which were situated mainly in Luzon. It is important to note, however, that this analysis was based on the local institutions that hold the copyright to the records, and collaborations with other local institutions were not considered.

Figure 6 Heatmap matrix showcasing the relationship between the number of barcode records associated with regions that have been sampled and the regions of local institutions that contributed the data.

There are officially seventeen regions in the Philippines, represented by the Philippine map (A), with non-numerical regions labelled as follows: ca, Cordillera Administrative Region (CAR); mm, National Capital Region (NCR or also referred to as Metro Manila); and br, Bangsamoro Autonomous Region in Muslim Mindanao (BARMM). Regions are also divided based on their island groups – namely Luzon (red), Visayas (yellow), and Mindanao (blue). For matrix B, contribution was based on the institution that holds the copyright to the image associated with the records. Regions along the x- and y-axis are sorted to provide spatial context, with the map as a reference. The diagonal line represents the “ideal” scenario wherein the region serving as the processing center of barcode data can sufficiently sample its own local area. This dataset includes the records with NA entries for country sampled.

Discussion

In this study, biodiversity records on animal and plant taxa found in the Philippines were systematically assessed by examining the extent of metadata gaps, taxonomic biases, and spatial biases in barcode data while using species occurrence data mainly as a baseline. Results show that much of the barcode data had missing information on records and publishing, geolocation, or taxonomic information. Moreover, it was observed that the amount of barcode data can be directly associated with the amount of species occurrence data available for a particular taxonomic group and sampling locality. Lastly, the results also reveal that majority of the barcode data came from foreign institutions and while local barcoding efforts have increased in the recent decades, much of it is due to Philippine institutions being based within Luzon.

Incompleteness of metadata in barcode data

Biodiversity records have been used in various fields of study to further understand the underlying processes that influence biodiversity. Barcode data, in particular, have broad applications in various fields–e.g., in understanding the processes affecting regions with high diversity (Crandall et al., 2019; Matias & Riginos, 2018), in assessing the quality and authenticity of food products sold in markets (Barbuto et al., 2010; Maralit et al., 2013; Pazartzi et al., 2019), in conservation (Deichmann et al., 2017), and in battling illegal wildlife trade (Hartvig et al., 2015). Despite the various uses of barcode data, its overall utility can be reflected by the completeness of its metadata. Publishing and records information, for instance, would be useful in finding relevant references for future research and examining the global, national, or local state of biodiversity documentation. For example, in a similar study that focused on animal barcoding in the Philippines, they found that only about 20% of records on native species could be traced back to local institutions (Fontanilla et al., 2014). With this kind of information, it would be easier to objectively assess the progress of a particular institution or country in contributing to DNA barcoding or, more generally, to biodiversity documentation. Additionally, while metadata may not directly contribute new knowledge on biodiversity and its processes, it can provide context on the records being generated–particularly in terms of who, when, and possibly why they were published for a particular taxon and/or locality. As previously discussed, many of the available barcode records have missing metadata. It might be possible to manually retrieve this information from journal publications linked to these records but when dealing with a large amount of data, this approach could become challenging.

Another example of highly useful metadata is geolocation. By providing this metadata, barcode records could then be used for studies that examine the role of geography in biodiversity–such as the case of biogeographic research. For example, existing barcode records made it possible to examine the processes behind the rich marine diversity in the Indo-Pacific region, particularly at the molecular level (Crandall et al., 2019; Matias & Riginos, 2018). These inferences would not have been possible without the information on the location where the specimens were collected. It is important to note that there is, however, a concern for accuracy when dealing with this kind of information. In this study, two kinds of geolocation information were encountered: the numerical coordinates and the descriptive information on the locality. Evidently, coordinates are relatively more accurate compared to descriptive information since they could be easily standardized and used in spatial analysis. However, most barcode records that were examined lacked coordinates. Contributors could have intentionally refrained from including such information in their records or restricted access to it in the database since coordinates–and geolocation in general–are considered to be “sensitive” data. Sensitive data is any kind of information that, if made public, would cause an ‘adverse effect’ (e.g., illegal or excessive collection, risk of disturbance) on the associated taxon or living individual (Chapman, 2020; Environmental Resources Information Network, 2016). Several governments–such as in Australia (Andrews, 2009; Environmental Resources Information Network, 2016) and Canada (AMEC Earth & Environmental, 2010)–have implemented legal policies that deal with sensitive information of vulnerable species (e.g., plants and sessile animals, threatened or rare species). These policies would then largely influence the guidelines of public databases–such as GBIF (Chapman, 2020)–on managing the accessibility of sensitive metadata. With many records lacking coordinates, the provinces pulled out from the descriptive information were utilized for the analysis. Descriptions of the locality could also be informative. However, this highly depends on how detailed and standardized they are which in turn, may depend on how familiar the contributors were with the names and administrative units associated with the areas being sampled. This may explain why the majority of entries with descriptive information (with or without coordinates) were relatively more difficult to parse through (see Fig. 1B), with some being unclear or inconclusive, while others were more informative.

While barcoding is a growing technique that has much potential in biodiversity studies, one of its more popular applications is in species identification (Hebert & Gregory, 2005). Thus, metadata on taxonomic information would prove essential for the DNA barcodes to be used as an effective database, particularly for applications where organisms are not sampled (i.e., environmental DNA). While the results show that many taxonomic groups (see Fig. 2) had incomplete taxonomic information or low species identification, they also identified potential taxa for further taxonomic studies. Additionally, as the knowledge on taxonomy and evolutionary relationships between different taxa grows, there is always a possibility for the classification of a particular taxon to change. For examples, minor and major revisions have recently been made in angiosperm (i.e., at the order and family levels) and annelid classification (i.e., whole evolutionary tree) (Chase et al., 2016; Zrzavý et al., 2009). These changes in the taxonomic classification may explain the anomalies observed in evaluating the percent of barcoded species, as represented by the NaN orders and families. Upon further inspection, these taxa mainly contained marine species, most of which were given the status of “Accepted” in the World Register of Marine Species (https://www.marinespecies.org/). Moreover, the barcode records associated with these NaN taxa were obtained specifically from BOLD. The current taxonomic metadata of these records may also need to be updated. However, it is unclear whether this responsibility falls with the contributors or the curators of the biodiversity data.

Overall, there were significant metadata gaps present in the current barcode records on Philippine biodiversity that were retrieved from GenBank and BOLD–particularly, the information on the sampling location and identity of the species. Regardless of whether these kinds of information are being collected by researchers, if they are not included in the submission to these public databases, they can be perceived as missing. In this study, due to the extent of missing information, not all barcode records were deemed useful in some of the analyses. This does not necessarily imply that barcode records with incomplete metadata are unusable but highlights how the completeness of metadata allows these records to be used in various kinds of analyses. Because of the importance of metadata, its collection and publication have been strongly advocated and have inspired the creation of a database for metadata (Deck et al., 2017). Thus, researchers and contributors need to acknowledge the importance of metadata and be aware that in order to increase the utility of current biodiversity records, there is a need to also increase the availability of metadata by collecting and properly sharing this information with public databases. With regards to sensitive data (e.g., coordinates of vulnerable species), it may be possible to acquire authorization from the contributors to access the metadata (Chapman, 2020). Otherwise, the sampling locality description may be a sufficient substitute for coordinates, provided that the entries are more standardized and informative up to the province level, at least.

Barcode data favoring commonly documented taxa

In examining for taxonomic biases, it was observed that the rate of barcoding of taxa was associated with how commonly they were observed (see Fig. 3). Given that species occurrence records are largely opportunistic in nature (Petersen et al., 2021), the strong association between species and genetic datasets may indicate certain biases that are inherent to barcode data as well. Other than commonness, other factors might contribute to the variability in barcoding effort across taxonomic groups in the Philippines. For example, popular research likely influenced interest in barcoding of specific taxonomic groups. These topics include high endemism of vertebrates and vascular plants, and the high marine biodiversity in the Philippines, which led to efforts of barcoding vertebrates, endemic plants, and reef fishes, respectively (Carpenter & Springer, 2005; Ong, 2002; Posa et al., 2008). The limited number of experts available in the Philippines could potentially contribute to the observed taxonomic bias (Arayata, 2019; Senate of the Philippines, 2017). This lack of expertise is evident, for example, in the online roster of experts provided by the Department of Environment and Natural Resources–Biodiversity Management Bureau (https://bmb.gov.ph/index.php/resources/roster-of-experts), where it is evident that not all plant and animal taxa are well-represented. Furthermore, in relation to the findings on spatial bias, most DNA barcoding is processed in institutions based in Metro Manila. Each of these universities has a limited number of researchers with research interest focused only on certain taxa. Although there may be local experts specialized in less-represented groups, these experts may be based in regions where there is limited access to molecular approaches. In this case, collaborations become essential in providing these experts access to molecular facilities. Due to these factors, more attention in Philippine barcoding may have been given to certain groups belonging to the following phyla/divisions: Chordata, Arthropoda, Mollusca, and Tracheophyta.

Some exceptions were observed from the general trend that high genetic data can be expected with high species data. For example, there is currently no barcode data for the Family Ceratobatrachidae (Phylum Chordata, Class Amphibia) despite more than 20 species of limestone-forest frogs (Platymantis) recorded in the Philippines (Siler et al., 2009). Given the high endemicity and potential cryptic species diversity among this group (Siler et al., 2009), DNA barcode data can prove valuable in documenting the diversity within this taxon. Among plants, the Family Dipterocarpaceae (Division Tracheophyta, Class Magnoliopsida) is an example of a taxon that lacks barcode data. This family contains ecologically important yet exploited and endangered tree species. Examples are species of the genus Parashorea, Shorea, and Hopea, which largely contribute to the tree diversity and richness in many Philippine forests such as Mt. Apo Natural Park and Rajah Sikatuna Protected Landscape (Aureo et al., 2020; Zapanta et al., 2019). Unfortunately, some species in this family have become vulnerable to exploitation brought by logging, leading to some being critically endangered (Aureo et al., 2020; Zapanta et al., 2019). The lack of barcode data for these animal and plant taxa translates to missed opportunities in obtaining valuable information for this group–information that could be used in understanding the diversity of these groups and in the conservation of vulnerable species.

Barcode data favoring areas with high species documentation & foreign contributors

In examining for spatial biases, a similar trend was observed with the taxonomic biases. Specifically, examination of location information showed that barcode sampling is more likely conducted in areas where documentation of species is commonly done (see Fig. 4). The results revealed that the five provinces with the highest barcode sampling were Siquijor, Cavite, Bohol, Aurora, and Palawan. Three of these provinces are found in Luzon making them accessible to institutes that had the capacity to barcode. This accessibility however does not only pertain to proximity to barcoding institutions, but also to protected areas as well as the availability of routes to sampling locations (Fisher-Phelps et al., 2017; Oliveira et al., 2016). Indeed, in the Philippines, local biodiversity more frequently sampled are situated in provinces with more developed travel routes (or relatively near to urban areas). Security and safety are also linked to accessibility of an area. Governments often provide travel advisories that restrict access to certain areas due to the high risk of threats such as disease outbreaks, natural disasters, civil unrest, or terrorist attacks (Foreign, Commonwealth & Development Office, 2013). For instance, foreign researchers, who have been observed to generate a large portion of Philippine barcode data, are often strongly advised against travelling to many provinces in Mindanao due to “crime, terrorism, civil unrest and kidnapping” (Government of Canada, 2021; U.S. Department of State, 2021). As a result, provinces that are deemed to have lower risks to local and foreign researchers are more likely to be sampled compared to other provinces.

Another aspect of spatial bias examined in this study was the origins of contributors. It must be noted, however, that in this study, institutions holding the image copyright (specifically for BOLD entries with images associated with them) were assumed to be the submitter of the barcode data. In contrast to BOLD, submitter information is more explicitly indicated in GenBank entries. From a global perspective, most of the current barcode data of Philippine biodiversity was generated by foreign institutions with researchers from the United States being the most active contributors (see Fig. 5A). The high contribution of foreign institutions is likely due to their high research capacity, especially in terms of funding and barcoding facilities. For example, there exists a grant known as the “PIRE: Centennial Genetic and Species Transformations in the Epicenter of Marine Biodiversity” that enables researchers from various institutes based in the United States to conduct marine expeditions in the Philippines (Carpenter et al., 2017). Moreover, foreign institutions may also have access to more extensive specimen collections. For example, the Smithsonian National Museum of Natural History houses over 126 million specimens in their catalog. Additionally, the United States has about 1,500 other institutions that may also house a significant number of cataloged specimens but often with restricted access (Page et al., 2015). It is likely that many of their specimens, not exclusive to the United States, had been sampled even during the early years of exploration, which may explain why there are several barcode records generated from older samples.

Examination of contribution to barcode data across time showed that Philippines has become more active in barcoding in recent decades, particularly in terms of collecting samples and submitting barcode data (see Figs. 5B and 5C). The upward trend in both collection of samples and submission of barcode data seemed to have started between 2005 and 2010, around the time DNA barcoding was slowly being adopted in the Philippines. For example, the UP-Diliman Institute of Biology initiated the creation of a public DNA barcode database in 2008 and several years later, partnered with the Department of Environment and Natural Resources, to use DNA barcoding against illegal wildlife trade (Encarnacion, 2019).

While local contributions to barcode data have increased over the years, spatial bias was still prominent when the origins of contributors were examined from a national perspective. Specifically, there was a mismatch between the localities producing (or processing) the barcode data and the areas that were being sampled. This mismatch is likely due to the limited number of local institutions with the capacity to process and generate barcode data, whether in terms of facilities, funding, equipment, or expertise. Most of the local contributions are processed by a small group of institutions located in the regions of Metro Manila and Central Luzon (see Fig. 6)–many of which, if not all, have their own well-equipped DNA barcode laboratories. In line with this, it may be possible to increase the capacity of local institutions found in regions where there is currently minimal to no processing of barcode data by establishing the appropriate facilities and conducting professional training. While this will require funding and time, it could empower more local institutions to take initiative in barcoding their own local biodiversity–particularly those based in regions that remain relatively unexplored. This would be ideal as these local institutions are in the best position to sample their local biodiversity. Alternatively, collaborations with other local institutions (e.g., local government agencies, non-governmental organizations, etc.) can facilitate barcoding of local biodiversity. Indeed, many of the current barcode records are a product of collaborations between institutions based in Metro Manila and various local groups across the Philippines. While these may be indicated in the publications linked to these records, there is no clear metadata information on collaborative works provided on the raw barcode data obtained. The present limitation in the contributor metadata of these public databases potentially under-represents the role of local institutions in the documentation of Philippine biodiversity. For barcoding in particular, it is essential to acknowledge that both sampling and barcode generating efforts are equally important. Hence, institutions who contributed to either or both efforts in collaborations must also be credited equally–whether in publications or databases. Thus, a more explicit acknowledgment of the roles of local collaborators in the metadata associated with barcode data would increase the visibility of these local institutes, which could potentially foster further collaborations in biodiversity documentation.

Conclusions

By conducting a systematic assessment of the barcode data on animal and plant taxa, the state of barcoding in the Philippines was examined, giving insight on the extent of metadata gaps, taxonomic biases, and spatial biases present in current records. In analyzing the data, many barcode records were found to have missing information for publishing, records, geolocation, or taxonomic metadata. These gaps resulted in the exclusion of those records in some of the analyses, demonstrating that incompleteness of metadata can limit the usability of barcode data for different kinds of analyses. Also, the presence of metadata gaps makes biodiversity data more tedious to work with. Philippine barcoding is more often conducted on taxa and provinces that are associated with high documentation of species occurrence, with most records generated by foreign countries with generally high research capacity. Moving forward with the findings of this study, future contributors of barcode data are encouraged to increase the availability of metadata by collecting and sharing this information to online databases upon submission to maximize the potential utility of these records in various kinds of analyses. Additionally, future barcoding efforts should prioritize areas where biodiversity documentation is currently lacking such as documenting taxa and sampling regions that are under-represented in Philippine biodiversity data. This approach of sampling under-represented taxa and regions may be done by collaborating with institutions active in DNA barcoding and biodiversity experts specializing in less-represented animal or plant taxa and by conducting field sampling in locations that currently have limited data. Furthermore, it is essential to highlight the importance of empowering more local institutions to take part in Philippine barcoding whether by increasing their capacity to generate barcode data or collaborating with groups from different regions in the Philippines. For future studies on the biases and gaps in biodiversity data, collaborations with data scientists are also recommended to mitigate the tedious work involved in processing large amounts of data.

Supplemental Information

Supplemental Information 1 Supplementary Materials.

This file contains the different the supplementary tables for the manuscript and the supplementary figures, which are the same figures presented in the main text but using different data.

Click here for additional data file.

We would like to acknowledge and thank the UP Institute of Biology–with special mention of Sir Adrian U. Luczon–for providing support through feedback and consultation.

Additional Information and Declarations

Competing Interests

Author Contributions

Data Availability

The authors declare that they have no competing interests.

Carmela Maria P. Berba performed the experiments, analyzed the data, prepared figures and/or tables, authored or reviewed drafts of the paper, and approved the final draft.

Ambrocio Melvin A. Matias conceived and designed the experiments, performed the experiments, analyzed the data, prepared figures and/or tables, authored or reviewed drafts of the paper, and approved the final draft.

The following information was supplied regarding data availability:

The Rscripts used for the analysis and preparation of the figures are available at GitHub: https://github.com/miaberba/2021_PH_BiodiversityAssessment and https://github.com/dinmatias/GeneBankParse.

The data inputs for the Rscripts are available at Zenodo: Berba, Carmela Maria P., & Matias, Ambrocio Melvin A. (2022). State of biodiversity documentation in the Philippines: Metadata gaps, taxonomic biases, and spatial biases in the DNA barcode data of animal and plant taxa in the context of species occurrence data [Data set]. Zenodo. https://doi.org/10.5281/zenodo.6153441.

All data used in the analyses are publicly available at: (1) GenBank and Barcode of Life Data System for barcode data; (2) Global Biodiversity Information Facility for species occurrence data; (3) Philippines Statistics Authority (PSA) for the Philippine Standard Geographic Code database; and (4) Humanitarian Data Exchange of the United Nations Office for the Coordination of Humanitarian Affairs Office for the Philippines-Subnational Administrative Boundaries, originally sourced from the PSA and National Mapping and Resource Information Authority.

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
