# Peer review of "State of biodiversity documentation in the Philippines: Metadata gaps, taxonomic biases, and spatial biases in the DNA barcode data of animal and plant taxa in the context of species occurrence data"

_PeerJ, doi:10.7717/peerj.13146_

## Round 0.1 · original submission · Major Revisions

I agree with comments of reviewer 3 regarding the originality of a metadata paper like this article. More discussion is needed to clarify the status and the usefulness of the incomplete metadata of barcodes for the biodiversity in the Philippines and propose strategies to complete barcodes for certain groups of organisms. In addition, classification of plants should be reviewed to properly assign phylum. Also, it is recommended a review of the English to reach a professional level.

Reviewer 1 ·

Basic reporting

ok

Experimental design

Ok

Validity of the findings

ok

Additional comments

The paper is very interesting and enlightening. It has identified the gaps and clearly led readers and workers on the field the rigt research direction to take. I recommend to publish the paper as it is now.

Reviewer 2 ·

Basic reporting

This is a well-written review paper about the current state of the biodiversity documentation in the Philippines. However, I have some minor comments on the manuscript:

Lines 39-152: There was no mention regarding to the estimated number of species of plants and animals in the Philippines. I think there is a need to mention about this in the introduction as baseline information. The authors may add references about the richness of the flora and fauna in the Philippines.

Line 226: “data on Homo sapiens and Homo luzonensis” change to “data on Homo sapiens and H. luzonensis”

Experimental design

This paper utilizes the available data from the different online data-bases and it was properly acknowledged.

Validity of the findings

The results are well supported by the presented data.

Figures are clear and well-prepared.

Lines 337-338- The data on the coordinates are not usually indicated in the publication (for plants) because of some conservation issues. You may cite references about these issues, like it will encourage the poachers to collect the specimens etc.

Additional comments

This paper focuses on the review about the state of biodiversity documentation in the Philippines. However, this paper did not mention anything about the background on the biodiversity of the Philippines. The authors can check the book of Catibog-Sinha and Heaney (2013). It is a very important reference about the biodiversity researches in the Philippines. I suggest that the authors should consider citing Catibog-Sinha and Heaney (2013).

Reviewer 3 ·

Basic reporting

This research work is interesting and can be seen as a contribution particularly in this so-called time of "big data".

The language in some parts of the manuscript can be improved with a bit of a technical tone, and by making the sentences more concise, non-repetitive, and lessening the use of conjunctions, or compound/complex sentences. Examples are: Lines 41-46, paragraph 2 of the Introduction.

For the reporting of the results and its discussion, the authors gave more emphasis/examples on animal taxa, while for floristic groups it is more in passing. Also, take note of the names of taxonomic groups: Marchantiophyta, while Tracheophyta is not a phylum. The presentation of the results, particularly in the section Initial processing of biodiversity data, will be better if the authors provided a Table instead of a histogram to show details of taxonomic groups (higher level), number of taxa, genetic marker, number of records, etc. to really show the available data. As it is not unfamiliar to barcoders, that COI is mostly utilized for animal groups, while matK and rbcL are for plants, and one can assume that the reported frequency for COI is just because of the high number of available barcodes for animals that were used for this study. See Lines 326-327. For figures, take into consideration the readers who are not from the Philippines (Fig 6A), what does ca and mm mean?

Experimental design

Is this considered a systematic review, or falls under meta-analysis?

Although the need situation is well elucidated, the gaps presented are not new nor exclusive to the Philippines (e.g.: over-representation of taxa, concerns on spatial data, incomplete metadata of barcodes), as these are "personally" well-noted by most workers dealing with genetic/barcode data. To add novelty and relevance to this paper, I suggest that the authors provide/suggest a well-thought system/tool that workers can use/adopt when handling metadata prior to their submission of sequences to different databases.

The protocols presented in the Materials and Methods portion are repeatable. However, in my opinion, it is better to present the protocols with a scientific tone rather than "everyday written English".

Validity of the findings

As mentioned above, the results may not be novel anymore, or it appears that it was just repackaged as "in the Philippine setting", as the points being raised are already recognized by the community (but not often in written form). The presentation of figures/graphs is insightful in some aspects to give points to the arguments/discussion, but there are things that the authors should have considered before comparing/ coming up with "conclusions". Examples: Taxonomic biases in Philippine barcode data= In lines 375-377 it was mentioned that barcode records are related to the taxonomic diversity of a group. Definitely, taxa with a high number of representatives will really have a high barcode record compared with taxa with a low number of species just like what was mentioned in line 382 (phylum Marchantiophyta). In addition, one of the goals of barcoding is to unravel the diversity (incl. taxonomic diversity) within a group of organisms, and large taxa receive greater attention when it comes to DNA barcoding than smaller groups as the circumscription of taxa in the latter group is more often clear-cut. For spatial biases in Philippine barcode data, does the location/accessibility to these provinces play a factor, as well as forest cover when it comes to terrestrial organisms? For Barcode data favoring commonly documented taxa, is it only the level of endemism that plays a factor, or does it include the number of Philippine workers who deal with these taxa? There are a lot of points that the authors should consider when it comes to discussing their results, as the findings raised key points that have multiple root cause; hence, should be viewed with different lenses and supported by literature.

The conclusion part (lines 625-627) claimed that the study elucidated that incomplete metadata of barcodes can limit its useability, but this was not well discussed in the text. Also, some parts of the conclusion appeared like raising a note for action, it might be better to provide a system/tool/pipeline on how Philippine workers can mitigate the problem, as the authors mentioned in the text big data can be a tedious

Additional comments

NA

---

## Round 0.2 · Minor Revisions

The reviewer considers that the paper was much improved, however, they mentioned as well that the English of the article can be improved to reach a professional level. I encourage you to use either an editorial English company or find a proficient English speaker to edit the paper.

Reviewer 3 ·

Basic reporting

The current submission exhibited significant improvements. Language can be improved during copy editing.

Experimental design

Methods/protocols are clear.

Validity of the findings

The reported findings, although not that novel, presented a call to action not just for Philippine workers but for all who are working with biodiversity/DNA-based data.

---

## Round 0.3 · accepted · Accept

Thank you for reviewing the English, the manuscript is much more readable now. Also, the addition of the section on Data accessibility is very useful for readers and gives more strength to your paper.